# Eggplant Flour as a Functional Ingredient in Frankfurt-Type Sausages: Design, Preparation and Evaluation

**DOI:** 10.3390/foods14040624

**Published:** 2025-02-13

**Authors:** Jenny Blumenthal-Rodriguez, Carlos A. Amaya-Guerra, Armando Quintero-Ramos, Sandra L. Castillo-Hernández, Minerva Bautista-Villarreal, Juan G. Báez-González, Joel H. Elizondo-Luevano, Cynthia Torres-Alvarez

**Affiliations:** 1Universidad Autónoma de Nuevo León, Facultad de Ciencias Biológicas, Departamento de Alimentos, Av. Universidad S/N, Cd. Universitaria, San Nicolás de los Garza 66450, Mexico; jennyblumenthal@outlook.de (J.B.-R.); sandra.castilloh@uanl.mx (S.L.C.-H.); minerva.bautistavl@uanl.edu.mx (M.B.-V.); juan.baezgn@uanl.edu.mx (J.G.B.-G.); joel.elizondolv@uanl.edu.mx (J.H.E.-L.); 2Universidad ISA, Departamento de Ingeniería en Tecnología de Alimentos, Av. Presidente Antonio Guzmán Fernández Km. 5, Santiago de los Caballeros 51011, Dominican Republic; 3Universidad Autónoma de Chihuahua, Facultad de Ciencias Químicas, Campus Universitario #2, Circuito Universitario s/n., Chihuahua 31125, Mexico; aquinter@uach.mx; 4Universidad Autónoma de Nuevo León, Facultad de Agronomía, Francisco Villa S/N, Col. Ex-Hacienda El Canadá, General Escobedo 66050, Mexico

**Keywords:** eggplant flour, functional ingredient, sausages

## Abstract

The food industry faces significant challenges in meeting the demand for healthier and more sustainable products, particularly in the search for natural alternatives that optimize the nutritional value of processed products and, at the same time, improve their functionality and shelf life. The objective of this study was to evaluate eggplant flour (EF) in different concentrations (2–9%) as an alternative ingredient in the production of Frankfurt sausages. The physicochemical (Aw and pH) and chromatic properties, as well as the antioxidant activity (ABTS, DPPH, ORAC), total phenolic content, flavonoids, and sensory evaluation were analyzed. The physicochemical properties were as follows: Aw showed an average value of 0.973, and the average pH was 6.35. Additionally, significant differences were observed in chromatic properties across treatments. The antioxidant activity values ranged between 586.4–1775.5 ABTS µM eq. of Trolox/g of sausage and 550.7–1764.0 µM eq. of Trolox/g of sausage, indicating an improvement in activity as EF concentration increased. Treatments containing EF at 2% (T2) and 3% (T3) achieved an average acceptability rating close to 4 (“I like it slightly”), demonstrating a balance between functionality and sensory acceptability, although higher concentrations adversely affected flavor and texture perception. Eggplant flour was an effective and sustainable choice for enhancing meat products’ quality and shelf life, contributing to the development of healthier and more sustainable food options.

## 1. Introduction

The growing demand for healthier and more sustainable products has driven the food industry to seek natural alternatives that enhance nutritional value while improving product functionality and shelf life. By 2050, global agricultural production is expected to increase by 70%, necessitating that food systems transform to minimize environmental impact while ensuring access to adequate nutrition. One promising approach is the development of hybrid products that combine plant-based ingredients with meat, aiming to reduce overall meat consumption and promote higher vegetable intake [1]. The global vegan food market is projected to reach $37.5 million by 2030, with an anticipated compound annual growth rate (CAGR) of 10.7% between 2023 and 2030 [2]. There is a growing trend of incorporating functional ingredients of plant origin, such as legume flours, into meat products to improve their texture and nutritional value [3].

Recently, eggplant (*Solanum melongena* L.) has been shown to have great value because of its phytochemical composition, in particular its content of dietary fiber and polyphenols; as a result, it is considered a nutraceutical [4,5]. Eggplant is an economically important crop in tropical and subtropical regions of the world [4], and it is characterized by fruits of various sizes, shapes, and colors [6]. In 2022, global production exceeded 59 billion tons, with China and India as the primary producers [7]. In 2023, production in Mexico surpassed 91 million tons, with Sinaloa being the leading state in production; however, it is not widely utilized in the country [8]. Mexico ranks as the third-largest exporter of this vegetable to countries such as the United States, Germany, and the United Kingdom [9]. Previous research has highlighted the antioxidant properties of eggplant flour and its high content of dietary fiber. The significant concentrations of phenolic compounds in the eggplant skin are responsible for these properties. In this regard, previous studies by Murakami et al. [10] and Niño-Medina et al. [6] confirm its ability to reduce oxidation and improve the stability of foods containing it.

In meat products, lipid and protein oxidation are inevitable processes that negatively affect flavor and nutritional value, so using natural antioxidants from plant sources is a viable solution [11]. In addition, eggplant’s bioactive properties, attributed to the presence of phenolic compounds such as chlorogenic acid (in the fruit pulp) and anthocyanins (in the fruit skin), offer various health benefits and are currently being actively researched as potential treatments for metabolic and cardiovascular diseases [12,13]. Sembiring & Chin [11] evaluated the effect of eggplant powder on the quality of sausages during 35 days of refrigerated storage. The results indicated a decrease in the redness values (a*) and an increase in the yellowness values (b*) of the product; in addition, they observed lower levels of thiobarbituric acid (TBARS), suggesting antioxidant activity. Eggplant powder also inhibited microbial growth, extending the shelf life of sausages under refrigeration. These findings support the potential of eggplant powder as an antioxidant and antimicrobial agent in meat products.

Furthermore, Zhu et al. [14] investigated the addition of eggplant powder to sausages with soybean oil; they found that this addition significantly improved their textural and sensory properties while reducing fat and cholesterol and concluded that 2% eggplant powder was the most effective option, enhancing the flavor and overall acceptability of the product. These findings promote the use of eggplant powder in producing healthier sausages. Bunmee et al. [15] studied eggplant flour (EF) as a fat substitute in beef burgers. When they added EF, they observed increased moisture and fiber and decreased protein and fat. Burgers with EF showed better texture, juiciness, and stability during storage due to the antioxidant properties, and the authors concluded that EF can be a healthy and functional alternative for the production of burgers. As the authors aim to make a scientific contribution to food development, particularly in the meat industry, the main objective was to evaluate the impact of different concentrations of eggplant flour on the physicochemical, antioxidant, and sensory properties of Frankfurt-type sausages. In addition, formulations with and without phosphates were compared to explore the interactions between flour and traditional preservatives.

## 2. Materials and Methods

### 2.1. Flour Preparation

The eggplant *Solanum melongena* var. *esculentum* fruits used in this study were purchased at the local market in San Nicolás de los Garza, Nuevo León, Mexico. The fruit was washed and disinfected, then sorted for processing into four groups: whole eggplant (EF), eggplant peel (PL), eggplant external pulp (EP), and eggplant internal pulp (IP). The fruit was cut into slices and dehydrated at 40 °C to 45 °C for 16 h using a tunnel dryer (Procmex Model LQ001, Procomm, Monterrey, Mexico) [4,16]. The selected drying-temperature range was determined based on the findings of Uthumporn et al. and Vega-Galvez et al. [4,16] to preserve the bioactive compounds found in eggplant [17]. The particle size was reduced to 80 microns using a Cyclone Model 3010-030 (Fort Collins, CO, USA) sample mill. The flours were vacuum-packed and stored at room temperature in the dark until use. Flour yield was calculated using Equation (1), as follows:(1)$$Yield\;(\%)=\left(\frac{W1}{W0}\right)\times 100\;$$
where

*W*1 = finished weight of flour

*W*0 = fresh weight of cereal or vegetable

### 2.2. Functional Properties

Functional properties were determined using the method of Rodriguez-Jimenez et al. [17]. To assess water-holding capacity, 5 mL of water was adjusted to pH 7 and mixed with 0.5 g of a sample on a vortex mixer (Boeco, model V2H, Hamburg, Germany) for 1 min. The mixture was then centrifuged for 30 min at a speed of 3000 rpm. Finally, the water retention was calculated per gram of sample. Oil-retention capacity was assessed using a similar method, exchanging the water for vegetable oil. To assess emulsifying capacity, 5 g of the sample and 20 mL of distilled water were mixed, and the pH was adjusted. This mixture was processed in a vortex mixer for 15 min. Next, 20 mL of vegetable oil was added, and the mixture was homogenized (OMNI GLH model glh-01, OMNI International, Georgia, GE, USA) at medium speed for 3 min. The homogenized mixture was centrifuged at 1300 rpm. Results were expressed as the percentage relating the emulsion layer’s height to the total height of the liquid.

### 2.3. Total Dietary Fiber

The Total Dietary Fiber Assay Kit 111 (DF-100A and TDF-C10, Sigma-Aldrich, St. Louis, MA, USA) was used to measure total dietary fiber (TDF), along with its soluble (SDF) and insoluble (IDF) fractions. The analytical procedure adhered to the standardized methodology outlined in the Official Methods of Analysis [18].

### 2.4. Preparation of Flour Extracts (EFE)

A 90 mg portion of eggplant flour was extracted using 5 mL of 80% methanol. The extraction took place at room temperature with the help of a ceramic-top stirrer. After 40 min, each sample was centrifuged at 9500× *g* at 10 °C for 5 min [17]. The extract was separated from the sediment and kept cooled until further use.

### 2.5. Phenolic Compounds

The total phenol content was determined using the Folin−Ciocalteu method [19]. A mixture was prepared with 2.6 mL of distilled water, 0.2 mL of the test extract, 0.2 mL of the reagent (Folin−Ciocalteu), and 2 mL of a 7% sodium carbonate solution. After the sample was incubated for 2 h at room temperature (between 23 and 25 °C) while it was shielded from light, the absorbance was measured at 730 nm. The result was expressed as milligrams of chlorogenic acid equivalent (CEA) per 100 g of sausages or eggplant flour.

### 2.6. Total Flavonoid Content

The method described by López-Contreras et al. [20] was used with minor changes to measure total flavonoid content. A sample extract (200 μL) was combined with 3500 μL of distilled water and 150 μL sodium nitrite (NaNO_2_) at 5%. After 5 min, 150 μL of solution of 10% aluminum chloride (ALCl_3_) was added. The mixture was allowed to stand at room temperature, between 23 and 25 °C, and shielded from light for 5 min. Next, 1 mL of NaOH (1 M) was added, and the sample was mixed thoroughly for 5 s and set aside for 15 min under the same conditions. At 510 nm, the absorbance was measured. The result was expressed in milligrams of catechin equivalent (CAE) per 100 g of sausage or eggplant flour.

### 2.7. Preparation and Analysis of Frankfurt-Type Sausages

#### 2.7.1. Preparation of Raw Material 

The authors of this study combined lean beef and pork in a 1:1 ratio, adding 8% pork backfat to the total meat mixture. The weight of the meat-and-fat base was used to determine the proportions of condiments and additives. The experimental treatments were divided into two main groups. The EF Group was evaluated based on the percentage of eggplant flour substituted for the meat mix, with substitutions at 2%, 3%, and 5%—the lower and upper limits set by the Mexican Standard NMX-F-065-1984 [21]—as well as at 9%, a value exceeding the recommended limits, to investigate the potential functional effects of the flour (Table 1). The second group included treatments with (+) or without (−) phosphates to test the impact of EF addition; for this purpose, the treatments with the best sensory attributes resulting from the previous group, T2 and T3, were considered (Table 2). Each group also had a corresponding control. The eggplant flour used in the experiment contained 12.7% protein, 12.17% fiber, 6.53% ash, 1.73% fat, and 65.22% carbohydrate [22].

#### 2.7.2. Preparation of the Frankfurt-Type Sausages

The order of addition to the cutter was as follows: meat, phosphates, salts, preservatives, water, protein extenders, flours, starches, water, and seasonings. The preparation was processed in batches of 10 kg, and the temperature was kept below 8 °C. The resulting product was termed “meat paste” [22]. The meat paste was manually stuffed into casings. The casings used were semipermeable synthetic casings, 26 gauge, which allowed heat to enter and exit the product. The stuffed sausages weighed approximately 16 to 20 g each. The stuffed product was cooked in a traditional box oven with a gas flame at the bottom. The cooking temperature of the oven varied from 80 to 90 °C. When the product reached a core temperature (using an NSF thermometer THDP-450 with a temperature range of −40 to 230 °C) of 72 °C, this was achieved in approximately 40 min. The cooking yield was calculated using Equation (2), as follows:(2)$$Yield\;(\%)=\left(\frac{W0}{W1}\right)\times 100\;\;$$
where

*W*1 = finished product weight

*W*0 = fresh weight of meat paste

### 2.8. pH and Water Activity (Aw)

For Frankfurt-type sausages, the pH was measured using a calibrated meat-specific pH meter (HANNA, model HI 99163, Woonsocket, RI, USA) by inserting a pH probe into three locations on each sausage. The water activity (a_w_) was determined using an Aqualab meter from Decagon Devices Inc. (Pullman, WA, USA). Three sausages were selected for each treatment to measure a_w_ and pH, with all analyses conducted in triplicate at ambient temperature.

### 2.9. Chromatic Parameters

To measuring color, a spectrophotometric cuvette was filled with a 1.5 mL sample and analyzed using a Konica Minolta (Tokyo, Japan) CR-20 color reader. Color measurements were determined using CIELAB (*L**, *a**, *b**) and CIELCH (L*, C*, h) Commission internationale de l’Éclairage [23]. The color representation was created with ColorHexa, an online color-conversion tool based on *L**, *a**, and *b** values [24]. In these systems, *L** represents lightness (ranging from 0 for black to 100 for white), *a** denotes red (positive) or green (negative), and *b** indicates yellow (positive) or blue (negative). Additionally, *C** corresponds to chroma (color intensity) and *h* represents the hue angle, with reference points at 0° (red), 90° (yellow), 180° (green), and 270° (blue).

### 2.10. Sensory Evaluation

The sensory analysis of the sausages was conducted using an affective test, specifically a hedonic test based on the general perception of five attributes: aroma, texture (bite), flavor, general appearance, and acceptability. Testing was conducted by a panel of 30 evaluators between 20 and 30 years old with a minimum of one semester of training in sensory evaluation and meat products. Panelists rated each attribute on a scale from 1 to 5, where (1) represents “very much disliked”, (2) represents “slightly disliked”, (3) represents “neither liked nor disliked”, (4) represents “liked slightly”, and (5) represents “liked a lot”. The sausages were served on individual plates and in pieces using a three-digit random code [25].

### 2.11. Moisture-Retention Test

The AOAC 925.15 [26] method was used to measure the moisture content of the Frankfurt-type sausages. Moisture content was assessed on days 0 and 5 of storage in refrigerated conditions. The results obtained were utilized to calculate the percentage loss or gain of moisture in the sausages relative to their initial weight.

### 2.12. Preparation of Sausage Extracts (EFE)

A total of 25 g of sausages was mixed with 50 mL of distilled water in a beaker covered with cling film. Then, the sausages were boiled for 20 min, cooled, and filtered using Whatman filter paper nº 4. The extract was held in a colored flask for immediate use [27].

### 2.13. Antioxidant Capacity

The eggplant-flour extract’s electron- and hydrogen-donating capacities were measured by ORAC, DPPH, and ABTS assays. For sausage, the last two methods were used. The DPPH method was performed with the technique described by Tai et al. [28], with slight modifications. In this technique, 1500 μL of DPPH solution (2 mg/L in 80% methanol) and 50 μL of sample were mixed and incubated in the dark for 30 min at room temperature (23–25 °C). The absorbance was measured at 517 nm. The findings were reported in micromoles of Trolox equivalents (μMTE) per gram of sausage or eggplant flour. The ABTS assay procedure followed the method used in previous assays [29,30], with some modifications. A stock solution was prepared from equal proportions 2.6 mM potassium persulfate and 7.7 mM of the reagent ABTS-+, dissolved in distilled water. Following 12 h at room temperature in the dark, the solution was diluted with 80% methanol to achieve an absorbance of 1000 units at 734 nm, as measured using the spectrophotometer. Next, 1.5 mL of ABTS-+ solution was allowed to react with 50 μL of extract. The absorbance at 734 nm was measured after 30 min of resting in the dark. Results are expressed as micromoles of Trolox equivalents (μMTE)/g of sausage or eggplant flour. The ORAC assay was performed using 96-well plates in an automated plate reader (Synergy 2, Bio Tek, Winooski, VA, USA). A pH 7.4 phosphate buffer was used. The radical was generated from 2, 2′-azobis (2-amidino-propane) prepared just before use, with fluorescein serving as the substrate. The readout range included excitation at 485 nm and emission at 520 nm [31]. The results, expressed as μM TE/g flour, were analyzed using Gen 5 software from BioTek (Winooski, VA, USA). A calibration curve with the Trolox standard was generated from 0 to 100 μM.

### 2.14. Statistical Analysis

The experimental data were analyzed for three replicates to evaluate the homogeneity of variances, with results expressed as means ± standard deviation (SD). Statistical comparisons were performed using one-way analysis of variance (ANOVA) and then Tukey’s post-hoc test in SPSS 17. A significance level of *p* < 0.05 was used to determine differences between means.

## 3. Results

### 3.1. Properties of Eggplant Flour

To determine the individual properties of the eggplant parts, flours were produced from their fractions: peel (PL), inner pulp (IP), outer pulp (EP), and whole eggplant (EF). The fresh yield from each eggplant fraction followed the order IP > EP > PL. The processing yields of the fractions and the whole eggplant followed the order EF > PL > IP > EP (Table 3); this was calculated based on the weight in grams of the fresh product. The processing yield obtained from IP and EP flours was significantly lower (*p* < 0.05) than that obtained from EF and PL flours. The drying performance of the feed correlated with its initial moisture content; in this study, we started with a feed with an initial moisture content of 93% [32]. Performance characteristics are crucial when producing new ingredients, as they ensure access to quality products at an affordable production cost [33].

The results for water-holding capacity (WHC) varied significantly (*p* < 0.05) among PL, IP, EP, and EF flours, ranging from 3.7 to 6.4 g water/g flour (Table 2). PL flour exhibited a WHC of 6.4 g water/g dry weight, the highest among all flours, while IP flour retained 3.7 g water/g dry weight. The significance of WHC lies in its ability to modify the texture and viscosity of formulated foods, making it significant in a diverse range of processed foods. Conversely, the results for oil-holding capacity (OHC) ranged from 2.03 to 2.27 g oil/g flour (Table 3). A study by Ukom et al. [34] indicated that EF had a high WHC (3.85 to 5.61 g/mg) and OHC (0.64 to 4.85 mg/mL) due to the pretreatments applied before flour processing; both WHC and OHC enhance sensory properties and, consequently, the product’s palatability when it is consumed. The flour’s emulsifying capacity (CE) results followed a trend similar to that seen in the OHC results. It was noted that the functional properties were linked to the dietary-fiber content of the flours. The total dietary fiber (TDF) of the flours was ranked as follows: PL > FL > EP > IP (Table 4). This pattern also appeared for insoluble dietary fiber (IDF) and water-holding capacity (WHC), although this relationship did not extend to OHC and CE.

Table 4 shows the results for soluble dietary fiber (SDF), insoluble dietary fiber (IDF), and total dietary fiber (TDF) in the flours. PL flour (52.74%) was found to have a significantly higher content (*p* < 0.05) than EF, EP, and IP flours. In EF, EP, and PL flours, TDF constituted the highest percentage of dietary fiber. The fiber results for eggplant flour agree with those reported by Murakami et al. [10], Uthumporn et al. [4], and Mirani, A. & Goli, M. [35]. Regarding the fractions, Mansoura, M. et al. [36] provided data comparable to those obtained in this research. The evaluated flours have a higher percentage of TDF compared to those typically used in the meat industry, such as whole wheat flour (14.4%) and wheat flour (3.7%) [37]. Thus, eggplant flour can be regarded as a functional ingredient due to its high content of insoluble dietary fiber, which enhances water retention in its structural matrix and creates low-viscosity mixtures. This insoluble-fiber content leads to an increase in fecal mass that accelerates intestinal transit [38], which is responsible for its role in reducing lipids and potentially preventing cardiovascular diseases [39].

The chromatic parameters of the flours are shown in Table 5. This table shows the brightness (L), redness-greenness (a*), yellowness-blueness (b*), chroma index, hue angle, and views of colors in flours. The PL sample presented a negative value for the a* parameter, and its L* parameter was the lowest among the flours; this flour was classified as “light gray”, while the EP, IP, and EF flours had colors classified as “beige”. The values found in this research are lower than those reported by Montoya-López et al. [40] for wheat flour (L* 92.01; a* 0.56; b* 9.78), EP flour was the sample with values closest to the chromatic values of wheat flour, followed by EF and IP flour. These color characteristics could present a challenge when incorporating flours into a food matrix with lighter tones, since they add color; conversely, they can be advantageous in products in which beige or slightly darker tones are desired.

It has been observed that there is a relationship between phenolic content and the color of eggplant, and this relationship is explained by the content of anthocyanin, which is primarily found in the peel. Table 6 presents the data on phenolic content, with the ranking PL > EF > IP > EP. The table displays the results for TPC, with the highest value being for PL flour (127.90 mg of chlorogenic acid/100 g of flour) and the lowest value being for EP flour (32.90 mg of chlorogenic acid/100 g). Concerning the results for flavonoid content (Table 6), a significant difference (*p* < 0.05) was found among the various flours, and the PE sample exhibited the lowest flavonoid content among the samples (49.44 mg Cat/100 g of flour).

The results for the antioxidant capacity of the flours differed significantly (*p* < 0.05). The ABTS assay results (Table 6) ranged from 1376.86 to 264.74 (µM Trolox equivalents/100 g flour), with the order being PL > EF > IP > EP. This trend was supported by the results obtained using the DPPH method (1293.33 to 129.84 µM Trolox equivalents/100 g flour) and ORAC (481.55 to 209.32 µM Trolox equivalents/100 g flour). Niño-Medina et al. [6] reported similar findings: 785 µM Trolox equivalents/100 g for freeze-dried American-type eggplant.

### 3.2. Properties of the Frankfurt-Type Sausages

#### 3.2.1. Physicochemical and Chromatic Properties of the Frankfurt-Type Sausages

Considering the properties and drying performance of eggplant flour and eggplant fractions, the Frankfurt-type sausage model was developed using whole-eggplant flour, allowing for the production of the flour without generating waste. The results of cooking performance are shown in Table 7 and show that performance was as follows: T2 > control > T3 > T5 > T9. Some differences in cooking performance were noted, with the T9 treatment differing from the control sausages, T2, T3, and T5. Generally, the tests with 2% and 3% addition levels yielded values closest to the values obtained for their respective controls. During the production process, it was noted that the sausage formulated with EF had greater volume than the control, though it did not weigh more. This may be attributed to the flour’s high content of insoluble fiber, which allows it to retain air and water in its matrix during pasta processing [38]. Conversely, considering the characteristics of the casings, which enable air to enter and exit, a sweating process can occur during cooling, significantly reducing the yield percentage.

A different behavior was for in the Aw measurements (Table 7) compared to the yield results, where formulations with a higher EF content showed higher Aw values. These results were anticipated because of the water-retention properties of eggplant flour. Apaza et al. [41], in their study on the inclusion of albumin, tara gum, and soy protein in a sausage made from llama meat, reported Aw values (0.979–0.981) and also mentioned that this property enhances the juiciness of the sausages. The pH values (Table 7) ranged from 6.34 to 6.37, corresponding to the percentage of EF added. The pH values were higher than those reported for Frankfurt sausages made with *Agaricus bisporus* and *Pleurotus ostreatus* flour [42], which fell within a range of 5.94 to 6.11. It is important to clarify that the formulations in both the research by Cerón-Guevara et al. [42] and this study included phosphates (0.5% and 0.45%, respectively). Including phosphates in meat formulations can increase pH by 0.05 to 0.3 units in cooked products and by 0.1 to 0.7 pH units in raw-meat products [43]. Thus, their addition enhances the water-retention capacity of Frankfurt-type sausages.

The values of the color coordinates (L*, a*, and b*) of the sausages enhanced with EF, along with the values of saturation (chroma), hue (Hº), and difference (ΔE*) relative to the control, are presented in Table 7. The results for L* ranged from 58.92 to 32.43 for the control and the sausage with 9% EF, respectively. It was noted that incorporating eggplant flour into the meat matrix significantly reduced the parameter L*. This may be influenced by the temperature changes to which the sausages are subjected, which in turn trigger non-enzymatic darkening reactions of the carbohydrates, resulting in a range of colors from light yellow to dark brown [44] and the chromatic properties of eggplant flour. The values of the parameters a* and b* were found to lie in the ranges 11.27 to 13.12 and 17.65 to 18.38, respectively. Warris et al. [45] established a correlation between parameter a* and the iron content and the presence of heme pigments in the meat; as the iron and heme pigment content increase, the parameter a* also rises. This relationship was unaffected by the addition of flour. Conversely, a higher percentage of eggplant flour results in a lower b value. Some studies indicate that the b* value is influenced by the characteristics of the fat in the meat and is associated with the content of oxidized nitrosomoglobin; this pigment contributes to yellow hues in the meat [46,47].

All treatments showed significant differences compared to the control (*p* < 0.05), with these differences becoming more apparent with increased EF content. The data show that treatment 9 had a more substantial visual impact, as evidenced by the views (Table 7). The values for the color-saturation parameter, chroma, were significantly lower (*p* < 0.05) for treatment 9 than for the control (17.62). Regarding the results of the position within the chromatic circle, hue, T9 had the smallest angle (20.73), indicating a more significant shift in color hue compared to the other treatments with the addition of EF. Using EF as a meat extender significantly affected the sausages’ visual appeal, shifting them away from the pink color typical of the control sample, as shown in Table 7.

#### 3.2.2. Sensory Characteristics and General Acceptability of the Frankfurt-Type Sausages

The sensory analysis of sausages formulated with eggplant flour (EF) obtained an average score of 3.25 on a scale where 5 corresponds to “I like it slightly”. Figure 1 presents the sensory results, highlighting that the T2 treatment resulted in attributes most similar to those of the control sausage, followed by T3, T5, and T9. The bitter and intense flavor of eggplant, attributed to the presence of phenolic compounds and tannins [17], was more pronounced in the formulations with a higher percentage of EF (Control: 4.14; T1: 3.9; T3: 3.72; T5: 3.14; T9: 2.24). Similarly, the texture of the sausages with a higher EF content presented less elasticity and firmness, moving them away from the characteristic bite of the Frankfurt-type control product. Furthermore, the partial substitution of meat with EF hurt appearance due to the brown tone this ingredient confers, significantly reducing the acceptability and overall appearance scores. The results suggest that the higher the proportion of eggplant flour, the lower the preference of the panelists, a trend observed in all the sensory attributes evaluated. However, a high EF content is recommended in products such as pâtés, in which creaminess is favored, and the brown color is not a concern.

### 3.3. Sausage Moisture-Retention Test—Phosphate Test

Considering the influence of eggplant flour (EF) on water-retention capacity and pH, samples were formulated with and without the addition of phosphate to evaluate its effect and impact on these properties. To this end, analyses were conducted on the first day and after five days of storage to quantify moisture loss during this period, which corresponds to the production facility’s standard storage time. The results, presented in Table 8, include moisture-content values for both the initial day and day 5 of storage and variations in moisture throughout the storage process. On day 0, the moisture content of the samples with added phosphate was highest in the control sample, followed by the sample with ^+P^T3 and finally by the sample with ^+P^T2, which exhibited the lowest percentage. This pattern persisted during the first five days of storage for the phosphate-added samples. However, in the samples without phosphate, this behavior changed significantly: on day 5, the order was ^−P^T3 (60.77%), ^−P^T2 (59.01%), and Control (57.59%), with the control sample showing a noticeable moisture loss exceeding 6%. This difference in moisture loss can be attributed to eggplant flour’s (EF) water-retention capacity, which is the result of its significant content of dietary fiber, a component known for its ability to interact with water and influence the stability of food systems.

Table 9 shows the values of the color coordinates L*, a*, and b* for the sausage rind and its interior for formulations with or without phosphates, as well as the parameters of saturation (chroma), tone (Hº), and their difference (ΔE*) relative to the control. The luminosity values for the interior of the sausage with and without phosphate ranged from 52.3 to 59.26 and 48.61 to 58.89, respectively. The sausage rind presented lower values, from 37.59 to 46.12 (^+P^T3 and control, respectively) and 37.77 to 46.21(^−P^T3 and control, respectively). The noticeable difference between the rind and the interior of the sausage may be due to the use of high temperatures that trigger caramelization and Maillard reactions in the food matrix [4]. In addition to the above, the behavior of the samples in this research agree with the findings of Hernández and Güemes [48] in their study of sausage with the inclusion of orange peel; those authors found that the water content retained in the matrix directly affects the luminosity values.

The values of the a* coordinates for the sausage interior ranged from 8.94 to 9.07 and 8.91 to 9.15, with or without phosphate, respectively. The sausage rind presented lower values, from 11.12 to 11.69 (^+P^T3 and control, respectively) and 10.54 to 11.39 (^−P^T3 and control, respectively). For the chromatic coordinate b*, the sausage interior ranged from 15.02 to 18.40 and 15.13 to 18.27, with or without phosphate, respectively. The sausage rind presented lower values, from 16.91 to 18.26 (^+P^T3 and control, respectively) and 16.38 to 17.10 (^−P^T3 and control, respectively). Albarracín et al. [3], in their research on the effect of the use of bean flour as a meat extender, obtained values lower than those obtained in this research for the b* coordinate and higher values for the a* coordinate. The behavior of this last variable may have arisen because the proportions and type of meat (beef and pork) directly influence the values of the color coordinates, especially the a* coordinate. In this investigation, a 1:1 proportion of cow and pork meat was used; however, the investigation presented above used 100% beef; therefore, the values of the a* coordinate were high due to the iron content and the presence of heme pigments in the meat (Warris et al. [45]). The results for the color and visual appeal of the sausages (expressed in the coordinates L*, a*, and b*) are shown in Table 9.

The results obtained for the sausages’ antioxidant capacity showed a significant difference (*p* < 0.05) between the tests; the higher the EF content, the higher the antioxidant capacity (Figure 2). The ABTS values followed the order T2 > T1 > Control in the formulations with the addition of phosphates, as well as in the formulations without the addition of phosphates; the results varied from 642 to 1775 µM Trolox equivalents/g and from 586 to 1411 µM Trolox equivalents/g, respectively. The DPPH assay followed this trend (T2 > T1> Control), with values ranging from 500 to 1764 µM Trolox equivalents/g for sausages with and without added phosphates. The results show a trend that is to be expected when an ingredient with a high antioxidant content is added. Other authors have studied the use of antioxidants of natural origin from fruits and plant materials, finding that they have higher activity than synthetic antioxidants in different meat models [49,50,51,52,53]. The importance of the use of antioxidants in a meat matrix lies in the fact that they retard the oxidation of lipids and proteins [54], thus improving the quality and shelf life of meat products.

## 4. Conclusions

Eggplant has demonstrated significant health benefits and is emerging as a valuable ingredient in the food industry. This study found that eggplant-skin flour exhibited superior functional properties, including higher fiber content, higher antioxidant activity, and higher phenolic and flavonoid concentrations, relative to other flours. However, to promote sustainability and minimize waste, whole-eggplant flour was incorporated into Frankfurt-type sausages, leading to minimal changes in their physicochemical and chromatic properties. Sensory evaluation revealed higher consumer acceptance at lower concentrations (2–3%) than at higher levels (5–9%). Antioxidant activity increased proportionally with the addition of more eggplant flour, with the 3% formulation showing superior activity relative to the control. These findings support the potential of eggplant flour as a natural functional ingredient for enhancing the nutritional profile of processed foods while maintaining consumer acceptability. Further research is needed to explore its applications across different food matrices and thus optimize its use in innovative and nutritionally improved products.

## Figures and Tables

**Figure 1 foods-14-00624-f001:**
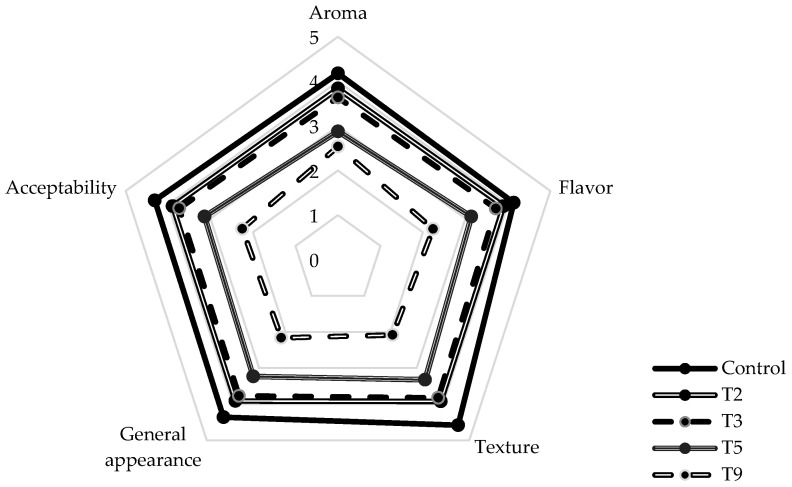
Sensory characteristics and acceptability of Frankfurt-type sausages with eggplant flour formulations.

**Figure 2 foods-14-00624-f002:**
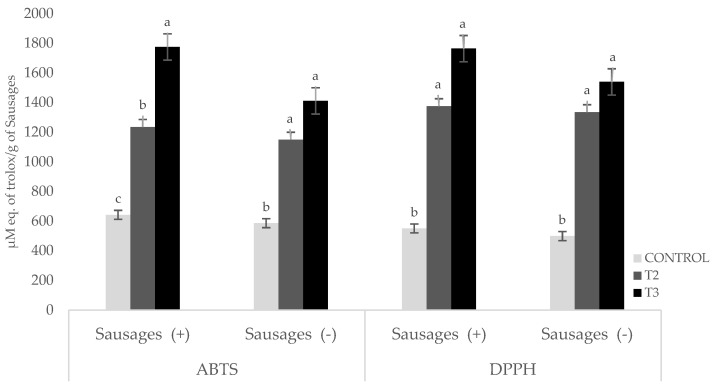
Antioxidant capacity of Frankfurt-type sausages, as measured by ABTS and DPPH. Sausages with phosphates (+) and/or phosphate-free sausages (−). Values are the average of three replicates. Values labeled with a different letter on the bars for samples with phosphates (+) or phosphate-free samples (−) are significantly different (*p* < 0.05).

**Table 1 foods-14-00624-t001:** Base formulation of Frankfurt-type sausages, EF group.

Ingredient	Percentage (%)				
	Control	T2	T3	T5	T9
Beef meat	33.5	32.58	32.12	31.21	29.36
Pork meat	33.5	32.58	32.12	31.21	29.36
Pork backfat	5.83	5.67	5.59	5.42	5.11
Eggplant flour	0	2	3	5	9
Vegetable protein	3.5	3.5	3.5	3.5	3.5
Water	18	18	18	18	18
Salts	2	2	2	2	2
* Preservatives	0.62	0.62	0.62	0.62	0.62
** Condiments	3.05	3.05	3.05	3.05	3.05

* Preservatives: phosphates, Sodium erythorbate.** Condiments: white pepper, sugar, mustard, nutmeg, monosodium glutamate, onion powder.

**Table 2 foods-14-00624-t002:** Base formulation of Frankfurt-type sausages, (+/−) phosphates group.

Ingredient	Percentage (%)					
	(+) Phosphates			(−) Phosphates		
	Control	^+P^T2	^+P^T3	Control	^−P^T2	^−P^T3
Beef Meat	33.50	32.58	32.12	33.71	32.58	32.12
Pork meat	33.50	32.58	32.12	33.71	32.58	32.12
Pork backfat	5.83	5.67	5.59	5.86	5.67	5.59
Eggplant flour	0	2	3	0	2.45	3.45
Phosphates	0.45	0.45	0.45	0	0	0
Vegetable protein	3.5	3.5	3.5	3.5	3.5	3.5
Water	18	18	18	18	18	18
Salts	2	2	2	2	2	2
* Preservatives	0.17	0.17	0.17	0.17	0.17	0.17
** Condiments	3.05	3.05	3.05	3.05	3.05	3.05

* Preservatives: phosphates, sodium erythorbate. ** Condiments: white pepper, sugar, mustard, nutmeg, monosodium glutamate, onion powder.

**Table 3 foods-14-00624-t003:** Yield and functional properties of eggplant flour and its fractions.

Flour	Yield (%)	WHC ^1^ (g Water/g Flour DW)	OHC ^2^ (g Oil/g Flour DW)	EC ^3^ (%)
EF	9.89 ± 0.3 ^a^	4.77 ± 0.2 ^b^	2.15 ± 0.1 ^b^	73.58 ± 1.3 ^b^
PL	9.82 ± 0.4 ^a^	6.44 ± 0.2 ^a^	2.27 ± 0.2 ^a^	86.79 ± 2.7 ^a^
IP	6.74 ± 0.3 ^b^	3.70 ± 0.1 ^c^	2.15 ± 0.1 ^ab^	66.78 ± 2.8 ^c^
EP	5.78 ± 0.1 ^b^	4.50 ± 0.07 ^b^	2.03 ± 0.2 ^b^	67.19 ± 2.3 ^c^

^1^ Water-holding capacity (WHC), ^2^ oil-holding capacity (OHC), and ^3^ emulsion capacity (EC) of eggplant flours and fractions. EF: eggplant flour, PL: eggplant peel flour, IP: eggplant internal pulp flour, EP: eggplant external pulp flour. DW: dried weight. The data represent the mean values from three replicates derived from three separate lots ± standard deviations. Within the same column, mean values marked with different letters indicate a statistically significant difference (*p* < 0.05).

**Table 4 foods-14-00624-t004:** Dietary-fiber content of eggplant flour and its fractions.

Flour	TDF (%)	IDF (%)	SDF (%)
EF	34.43 ± 0.1 ^b^	32.86 ± 0.3 ^ab^	1.57 ± 0.2 ^c^
PL	52.74 ± 0.2 ^a^	44.24 ± 0.1 ^a^	8.50 ± 0.1 ^b^
IP	25.97 ± 0.4 ^c^	8.77 ± 0.6 ^c^	17.20 ± 0.4 ^a^
EP	30.66 ± 0.8 ^b^	23.11 ± 0.2 ^b^	7.54 ± 0.8 ^b^

TDF: total dietary fiber, IDF: insoluble dietary fiber, SDF: soluble dietary fiber. EF: eggplant flour, PL: eggplant peel flour, IP: eggplant internal pulp flour, EP: eggplant external pulp flour. The data represent the mean values from three replicates ± standard deviations derived from three separate lots. Within the same column, mean values marked with different letters indicate a statistically significant difference (*p* < 0.05).

**Table 5 foods-14-00624-t005:** Chromatic properties of eggplant flour and its fractions.

Flour	Chromatic Properties					
	*L**	*a**	*b**	*h*	*C**	View
EF	69.1 ± 0.01 ^a^	1.8 ± 0.01 ^b^	18.2 ± 0.01 ^a^	84.3 ± 0.01 ^b^	18.3 ± 0.01 ^a^	
PL	57.1 ± 0.01 ^b^	−0.4 ± 0.1 ^b^	6.4 ± 0.01 ^b^	−86.6 ± 0.01 ^a^	6.4 ± 0.01 ^b^	
IP	68.6 ± 0.01 ^a^	4.3 ± 0.01 ^a^	22.6 ± 0.01 ^a^	79.2 ± 0.01 ^b^	22.9 ± 0.01 ^a^	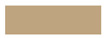
EP	72.2 ± 0.01 ^a^	1.9 ± 0.01 ^b^	23.2 ± 0.01 ^a^	85.2 ± 0.01 ^ab^	23.3 ± 0.01 ^a^	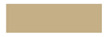

Chromatic properties (*L**, *a**, *b**, *C**, *h*) of eggplant flours, EF: eggplant flour, PL: eggplant peel flour, IP: eggplant internal pulp flour, EP: eggplant external pulp flour. The data represent the average values from three replicates obtained from three separate lots ± standard deviations. In the same column, mean values marked with different letters indicate a statistically significant difference (*p* < 0.05).

**Table 6 foods-14-00624-t006:** Antioxidant activity and phenolic compounds in eggplant flour and its fractions.

Flour	ABTS (μMTE/100 g Flour DW)	DPPH (μMTE/100 g Flour DW)	ORAC (μMTE/100 g Flour DW)	TPC ^1^ (mgCAE/100 g Flour DW)	TFC ^2^ (mgCatE/100 g Flour DW)
EF	629.39 ± 15.15 ^b^	526.66 ± 12.68 ^b^	354.69 ± 89.0 ^b^	80.26 ± 8.50 ^b^	161.66 ± 8.81 ^b^
PL	1376.86 ± 119.31 ^a^	1293.33 ± 155.91 ^a^	481.55 ± 90.0 ^a^	127.90 ± 6.45 ^a^	225.55 ± 4.631 ^a^
IP	462.72 ± 18.18 ^c^	420.31 ± 30.61 ^b^	232.75 ± 31.2 ^c^	67.76 ± 6.18 ^b^	175.00 ± 10.00 ^b^
EP	264.74 ± 17.75 ^d^	129.84 ± 26.22 ^c^	209.32 ± 75.1 ^c^	32.90 ± 2.83 ^c^	49.44 ± 5.09 ^c^

EF: eggplant flour, PL: eggplant peel flour, IP: eggplant internal pulp flour, EP: eggplant external pulp flour. ^1^ TPC: total phenols content, ^2^ TFC: total flavonoid content, mgCAE: milligrams of chlorogenic acid equivalents, mgCatE: milligrams of catechin equivalents, μMTE: micromoles Trolox equivalents. DW: dried weight. The data represent the mean values from three replicates obtained from three separate lots ± standard deviations. In the same column, mean values marked with different letters signify a statistically significant difference (*p* < 0.05).

**Table 7 foods-14-00624-t007:** Physicochemical and chromatic properties of the Frankfurt-type sausages.

Treatment	CP ^1^ (%)	Aw	pH	Chromatic Properties
*L**	*a**	*b**	Δ*E** ^2^	*C** ^3^	*Hº* ^4^	View
Control	98.11 ± 0.25 ^b^	0.9747 ± 0.62 ^a^	6.34 ± 0.15 ^a^	58.92 ± 0.03 ^a^	9.03 ± 0.01 ^e^	15.13 ± 0.03 ^d^	-	17.62 ± 0.03 ^a^	8.72 ± 0.01 ^e^	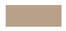
T2	98.43 ± 0.17 ^a^	0.9758 ± 0.23 ^a^	6.35 ± 0.17 ^a^	39.74 ± 0.03 ^b^	11.27 ± 0.01 ^d^	18.41 ± 0.03 ^a^	6.95 ± 0.02 ^d^	1.50 ± 0.01 ^cd^	83.89 ± 0.02 ^d^	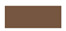
T3	97.48 ± 0.20 ^c^	0.9759 ± 0.60 ^a^	6.36 ± 0.08 ^a^	37.01 ± 0.01 ^c^	11.89 ± 0.02 ^c^	18.14 ± 0.03 ^b^	9.63 ± 0.02 ^c^	1.47 ± 0.01 ^cd^	58.73 ± 0.03 ^c^	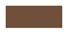
T5	97.43 ± 0.07 ^c^	0.9825 ± 0.15 ^b^	6.35 ± 0.28 ^a^	35.52 ± 0.02 ^d^	12.01 ± 0.03 ^b^	18.03 ± 0.02 ^c^	11.13 ± 0.03 ^b^	1.44 ± 0.01 ^de^	51.24 ± 0.02 ^b^	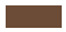
T9	96.18 ± 0.16 ^d^	0.9844 ± 0.60 ^b^	6.36 ± 0.60 ^a^	32.43 ± 0.01 ^e^	13.52 ± 0.03 ^a^	17.65 ± 0.01 ^e^	14.24 ± 0.01 ^a^	2.15 ± 0.03 ^b^	20.73± 0.02 ^a^	

CP ^1^: cooking performance; Aw: water activity; ^2^ ΔE*: color difference; ^3^ C*: chroma.; ^4^ H: hue. The data represent the mean values from three replicates derived from three separate lots ± standard deviations. Within the same column, mean values marked with different letters indicate a statistically significant difference (*p* < 0.05).

**Table 8 foods-14-00624-t008:** Moisture content versus time in Frankfurt-type sausages.

Treatment		Moisture (%)		Loss or Gain of Moisture (%)
		Day 0	Day 5	
With phosphates	Control	60.51 ± 0.10 ^a^	61.89 ± 0.20 ^a^	+2.28
	^+P^T2	59.31 ± 0.09 ^b^	59.80 ± 0.10 ^b^	+0.83
	^+P^T3	59.69 ± 0.30 ^b^	60.54 ± 0.20 ^ab^	+1.42
Phosphate-free	Control	61.39 ± 0.08 ^a^	57.59 ± 0.30 ^c^	−6.19
	^−P^T2	59.75 ± 0.12 ^b^	59.01 ± 0.20 ^b^	−1.24
	^−P^T3	60.52 ± 0.09 ^ab^	60.77 ± 0.20 ^a^	+0.41

The data represent the mean values from three replicates derived from three separate lots ± standard deviations. Within the same column, mean values marked with different letters indicate a statistically significant difference (*p* < 0.05).

**Table 9 foods-14-00624-t009:** Chromatic parameters of Frankfurt-type sausages.

Treatment	L*	a*	b*	ΔE* ^1^	C* ^2^	Hº ^3^	View
Sausages inside	With phosphats	Control	59.26 ± 0.1 ^a^	9.07 ± 0.01 ^a^	15.02 ± 0.0 ^c^	-	17.54 ± 0.0 ^c^	8.70 ± 0.04 ^c^	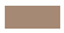
^+P^T2	52.69 ± 0.01 ^b^	8.94 ± 0.03 ^b^	17.88 ± 0.02 ^b^	7.16 ± 0.04 ^c^	19.99 ± 0.01 ^b^	9.63 ± 0.04 ^bc^	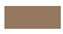
^+P^T3	52.3 ± 0.13 ^b^	9.07 ± 0.0 ^a^	18.40 ± 0.04 ^a^	7.71 ± 0.01 ^a^	20.51 ± 0.03 ^a^	9.83 ± 0.01 ^b^	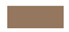
Phosphate free	Control	58.89 ± 0.02 ^a^	9.03 ± 0.0 ^b^	15.13 ± 0.01 ^c^	-	17.62 ± 0.02 ^b^	8.72 ± 0.04 ^c^	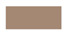
^−P^T2	51.64 ± 0.01 ^b^	8.91 ± 0.06 ^c^	18.27 ± 0.05 ^a^	7.90 ± 0.03 ^b^	20.33 ± 0.03 ^a^	9.79 ± 0.03 ^b^	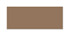
^−P^T3	48.61 ± 0.01 ^c^	9.15 ± 0.03 ^a^	17.95 ± 0.02 ^b^	10.66 ± 0.06 ^a^	20.14 ± 0.06 ^a^	10.66 ± 0.06 ^a^	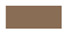
Sausages on the outside	With phosphats	Control	46.12 ± 0.01 ^a^	11.12 ± 0.01 ^c^	16.91 ± 0.03 ^c^	-	20.24 ± 0.06 ^b^	13.56 ± 0.03 ^d^	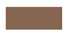
^+P^T2	39.75 ± 0.01 ^b^	11.29 ± 0.0 ^b^	17.24 ± 0.01 ^b^	6.38 ± 0.01 ^b^	20.61 ± 0.01 ^b^	15.86 ± 0.01 ^b^	
^+P^T3	37.59 ± 0.01 ^c^	11.69 ± 0.06 ^a^	18.26 ± 0.04 ^a^	8.66 ± 0.03 ^a^	21.68 ± 0.0 ^a^	17.27± 0.06 ^a^	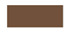
Phosphate free	Control	46.21 ± 0.03 ^a^	11.39 ± 0.04 ^a^	16.38 ± 0.02 ^c^	-	19.95 ± 0.0 ^b^	13.85 ± 0.04 ^b^	
^−P^T2	39.15 ± 0.04 ^b^	10.89 ± 0.03 ^ab^	17.42 ± 0.04 ^ab^	7.15 ± 0.04 ^b^	20.54 ± 0.01 ^a^	15.55 ± 0.03 ^a^	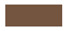
^−P^T3	37.77 ± 0.02 ^c^	10.54 ± 0.02 ^b^	17.10 ± 0.03 ^b^	8.51 ± 0.0 ^a^	20.08 ± 0.02 ^ab^	15.59 ± 0.06 ^a^	

^1^ ΔE*: color difference, ^2^ C*: chroma., ^3^ H: hue. The data represent the mean values from three replicates derived from three separate lots ± standard deviations. Within the same column, mean values marked with different letters indicate a statistically significant difference (*p* < 0.05).

## Data Availability

The original contributions presented in the study are included in the article; further inquiries can be directed to the corresponding authors.

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
