# Peer review of "Eggplant Flour as a Functional Ingredient in Frankfurt-Type Sausages: Design, Preparation and Evaluation"

_foods, 2025, doi:10.3390/foods14040624_

Round 1
Reviewer 1 Report
Comments and Suggestions for Authors
The article investigated the use of eggplant flour as an alternative ingredient in Frankfurt-Type production. The findings indicate that eggplant flour is an effective and sustainable option for improving the quality and shelf life of meat products, contributing to the development of healthier, more sustainable foods. However, some questions should be taken into further consideration for the integrity of the study.
1. I advise to add any numerical findings to the abstract.
2. Introduction section contains number of redundant lines about eggplant. I would suggest authors to optimize the language to make the content more concise.
3.There is no information why the percentage of eggplant meal from 2% to 9% was chosen.
4. In“Moisture Retention Test” section, why did choose days 0 and 5 of storage under refrigerated conditions to measure moisture content? Please add the rationale.
5. Discussions in support of the results sections are very shallow. I would suggest authors to strengthen the discussion section significantly.
6.The conclusion needed to be supplemented.
Comments on the Quality of English Language
The English could be improved to more clearly express the research.
Author Response
- I advise to add any numerical findings to the abstract.
Response: Thank you for your observation. Part of the abstract was redesigned by adding numerical data obtained in the study. We appreciate your feedback. Marked in blue, Lines 23-31.
- Introduction section contains number of redundant lines about eggplant. I would suggest authors to optimize the language to make the content more concise.
Response: Thank you for your observation. Some statements were removed. We appreciate your feedback.
- There is no information why the percentage of eggplant meal from 2% to 9% was chosen.
Response: Thank you for your comment. We have added an explanation regarding the selection of the eggplant meal percentage (2% to 9%) in the text. We appreciate your feedback. Marked in blue, Lines 152 – 155.
- In “Moisture Retention Test” section, why did choose days 0 and 5 of storage under refrigerated conditions to measure moisture content? Please add the rationale.
Response: Thank you for your comment. We have added the rationale for selecting days 0 and 5 of storage under refrigerated conditions to measure moisture content. We appreciate your feedback. Marked in blue, Lines 428-432.
- Discussions in support of the results sections are very shallow. I would suggest authors to strengthen the discussion section significantly.
Response: Thank you for your comment. We have strengthened the discussion section to provide a more in-depth analysis and better support the results. We appreciate your suggestion. Marked in blue, Lines 264 -266; 280-283; 294-295; 301- 302; 309–311; 316 – 319; 357 – 361; 369 – 373; 384 – 385; 389; 398 – 406; 413 – 423; 428 – 432; 436 – 440; 451 – 452; 454- 458; 466 – 472; 496- 497.
- The conclusion needed to be supplemented.
Response: Thank you for your valuable observation. We have added the concluding session in the manuscript to provide a more comprehensive summary of our findings. Marked in blue, Lines 505- 517.

Reviewer 2 Report
Comments and Suggestions for Authors
The manuscript deals with the properties of Eggplant Flour added to Frankfurt-Type sausages in different quantities. Eggplant flour has proven to be a good and sustainable alternative to improve the quality and shelf life of meat products. This would help to boost the production of healthier and more sustainable food. My comments and suggestions are given below:
Line 111: “The AOAC 985.29 [17] method was applied….”. In my opinion, it would be better to rewrite the sentence as for example: “Total Dietary Fiber Assay Kit 111 (DF-100A y TDF-C10 Sigma-Aldrich) was applied to determine the total dietary fiber (TDF), the soluble (SDF) and insoluble (IDF) fractions. The analytical procedure of the kit is based on the method published in the Official Methods of Analysis………”
Line 120: Folin-Ciocalteu method [18]. Deng-Cheng Liu et al (2009) [18] describe the method differently. Authors should check the method described in the article by Deng-Cheng Liu et al. (2009), and, if necessary, replace the reference.
Line 126: “Total flavonoids content”. The authors state that TFC was measured using the Xiong et al. [19] method, with slight modifications.
I have some questions about this, namely: a) the changes in the method seem substantial, rather than slight; b) on what principles did the authors make the changes to the method? c) Xiong et al. (2014) applied the method to edible flowers. I ask the authors if the changes of the method were necessary because the examined matrix was different?
Line 136:” lean beef was mixed with lean pork in a 1:1 ratio, and 8 % pork”. It would be better to specify that the percentage of lean beef and pork is given in Table 1.
Line 146: The meaning of the acronym NEF shall be specified
Line 147: Table 1 and Table 2 show, among other components, “Conservatives” and Condiments “. It would be better to specify what kind of products were used.
Lines 162-163: please include more information on the casing (e.g. the porosity and the manufacturer, if known)
Line 166: How was the core 72 °C temperature measured? How long did it take to cook the Frankfurt-Type Sausages? What were the dimensions of the sausages? It would be appropriate to specify these data.
Line 172: “Water Activity (Aw)”. It would be better “Activity Water (aw)”.
Line 205: “methods”. It would be better “assays”.
Line 268: “…..authors in eggplant.” One or more references should be written at the end of this sentence.
Lines 268-269: “In comparison………. wheat flour (3.7 %) [31]” This sentence is not clear enough.
The manuscript does not contain a title for the "Conclusions" section.
In the "Conclusions" section, authors should emphasize the concept of "sustainability", which they have referred above. Furthermore, the concept of "functional ingredient" for "eggplant flour" should be more closely related to the results obtained.
Finally, "Conclusions" calls for a better discussion of the shelf life of the products studied.
Comments on the Quality of English LanguageEnglish needs some improvements.
Author Response
- Line 111: “The AOAC 985.29 [17] method was applied….”. In my opinion, it would be better to rewrite the sentence as for example: “Total Dietary Fiber Assay Kit 111 (DF-100A y TDF-C10 Sigma-Aldrich) was applied to determine the total dietary fiber (TDF), the soluble (SDF) and insoluble (IDF) fractions. The analytical procedure of the kit is based on the method published in the Official Methods of Analysis………”
Response: Thank you very much for your comment. I have considered your suggestion to rewrite the sentence. We appreciate your feedback. Marked in blue, Lines 121- 124.
- Line 120: Folin-Ciocalteu method [18]. Deng-Cheng Liu et al (2009) [18] describe the method differently. Authors should check the method described in the article by Deng-Cheng Liu et al. (2009), and, if necessary, replace the reference.
Response: Thank you for your comment. We have reviewed the method description and the reference to Deng-Cheng Liu et al. (2009). It was a misunderstanding, and we have now corrected it accordingly. We appreciate your feedback. Marked in blue, Line 131.
- Line 126: “Total flavonoids content”. The authors state that TFC was measured using the Xiong et al. [19] method, with slight modifications. I have some questions about this, namely: a) the changes in the method seem substantial, rather than slight; b) on what principles did the authors make the changes to the method? c) Xiong et al. (2014) applied the method to edible flowers. I ask the authors if the changes of the method were necessary because the examined matrix was different?
Response: Thank you for your insightful comment. We have carefully reviewed the methodological description and have updated the citation to reflect the most appropriate reference. We appreciate your valuable feedback and have incorporated the necessary revisions accordingly. These changes are marked in blue on Lines 138.
- Line 136:” lean beef was mixed with lean pork in a 1:1 ratio, and 8 % pork”. It would be better to specify that the percentage of lean beef and pork is given in Table 1.
Response: Thank you for your comment. We have made the revision and now specify that the percentage of lean beef and pork is provided in Table 1. We appreciate your suggestion. Marked in blue, Lines 161 and- 172.
- Line 146: The meaning of the acronym NEF shall be specified.
Response: Thank you for your comment. We have changed and replaced "NEF" with "carbohydrate" for clarity. We appreciate your feedback. Marked in blue, Line 160.
- Line 147: Table 1 and Table 2 show, among other components, “Conservatives” and Condiments “. It would be better to specify what kind of products were used.
Response: Thank you for your comment. We have added the "Conservatives" and "Condiments" descriptions as a footnote in the tables to avoid making them more extensive. We appreciate your suggestion. Marked in blue, Lines 169- 170;172 – 173.
- Lines 162-163: please include more information on the casing (e.g. the porosity and the manufacturer, if known)
Response: Thank you for your comment. We have added the caliber of the casing as requested. However, we do not have information regarding its porosity or manufacturer. We appreciate your feedback. Marked in blue, Lines 179.
- Line 166: How was the core 72 °C temperature measured? How long did it take to cook the Frankfurt-Type Sausages? What were the dimensions of the sausages? It would be appropriate to specify these data.
Response: Thank you for your comment. We have added the details about the thermometer used to measure the core temperature. The cooking time is already specified in the document as approximately 40 minutes. The dimensions of the sausage correspond to the casing size, which is 26 gauge, added by one of your previous comments. However, we have no additional information beyond this. We appreciate your feedback. Marked in blue, Lines 182 -183.
- Line 172: “Water Activity (Aw)”. It would be better “Activity Water (aw)”.
Response: Thank you very much for your recommendation. However, in this case, we cannot accept it in this case, as the term "aw" is a standardized notation for water activity. We appreciate your feedback.
- Line 268: “…..authors in eggplant.” One or more references should be written at the end of this sentence.
Response: Thank you very much for your recommendation. The revision has been completed, and the ideas have been reorganized accordingly. Additionally, other authors have been cited to support the content. We appreciate your feedback.Marked in blue, Lines 294- 295.
- Lines 268-269: “In comparison………. wheat flour (3.7 %) [31]” This sentence is not clear enough. The manuscript does not contain a title for the "Conclusions" section.
Response: Thank you very much for your recommendation. The revision has been completed, and the ideas have been reorganized accordingly. We appreciate your feedback. Marked in blue, Lines 295- 297.
- In the "Conclusions" section, authors should emphasize the concept of "sustainability", which they have referred above. Furthermore, the concept of "functional ingredient" concept for "eggplant flour" should be more closely related to the results obtained.
Finally, "Conclusions" calls for a better discussion of the shelf life of the products studied.
Response: Thank you for your valuable feedback. We have strengthened the discussions to enhance clarity and coherence. Marked in blue, Lines 264 -266; 280-283; 294-295; 301- 302; 309–311; 316 – 319; 357 – 361; 369 – 373; 384 – 385; 389; 398 – 406; 413 – 423; 428 – 432; 436 – 440; 451 – 452; 454- 458; 466 – 472; 496- 497. A "Conclusions" section has also been incorporated to improve the manuscript’s structure. We appreciate your feedback. Marked in blue, Lines 505 - 517.

Reviewer 3 Report
Comments and Suggestions for Authors
This manuscript presents valuable research on the utilization of eggplant flour as a functional ingredient in sausages. With some additional discussion, clarification, and revisions, the manuscript could potentially make a significant contribution to the field of healthier and more sustainable meat product development.
1. The authors mention the growing trend of incorporating plant-based functional ingredients in meat products. Could they provide more context on the specific drivers behind this trend (e.g., consumer demand, regulatory changes, sustainability concerns) and how their research aligns with these drivers?
2. The authors could provide a more comprehensive literature review on the specific use of eggplant flour in sausages and related meat products.
3. While the authors highlight the antioxidant properties and fiber content of eggplant, they could elaborate on the potential mechanisms by which these properties could benefit meat products, particularly in terms of oxidative stability, shelf life, and texture.
4. In the flour preparation section, the authors mention drying the eggplant at 40-45°C to preserve phenolic compounds. Could they provide more details on the rationale behind this temperature range and its efficacy in retaining the desired compounds?
5. The authors should clarify the rationale behind the selection of specific eggplant flour concentrations (2%, 3%, 5%, and 9%) in the sausage formulations.
6. Additional information on the sensory evaluation protocol (e.g., sample preparation, serving order, panelist training) would be beneficial.
7. The authors mention using a "semi-trained" sensory panel. Could they provide more information on the training protocol, including the number of training sessions, the attributes evaluated, and the reference samples used for calibration?
8. The authors should provide a more in-depth discussion of the potential mechanisms behind the observed effects of eggplant flour on the physicochemical, antioxidant, and sensory properties of the sausages.
9. The authors observed a significant enhancement in antioxidant capacity and water retention in sausages with eggplant flour, particularly at 2-3% concentrations. Could they speculate on the potential mechanisms behind these effects and how they might contribute to improved shelf life and textural properties?
10. While higher concentrations of eggplant flour (5% and 9%) affected flavor and texture, could the authors provide more specific details on the nature of these sensory changes (e.g., off-flavors, toughness, dryness) and potential strategies to mitigate them?
11. The authors mention that eggplant flour proved to be an effective and sustainable option for improving meat product quality. Could they expand on the sustainability aspects of using eggplant flour, such as its environmental impact, resource efficiency, and potential for reducing food waste?
12. The discussion could be strengthened by comparing the findings with other relevant studies from the literature.
13. In the discussion section, the authors could compare their findings with other studies that have explored similar plant-based functional ingredients in meat products, highlighting the unique contributions of their research.
14. The authors could discuss the potential implications of their findings for other meat product categories beyond sausages, such as patties, meatballs, or restructured meat products.
15. It would be beneficial to provide recommendations for future research, such as exploring different processing methods, eggplant varieties, or eggplant flour combinations with other plant-based ingredients.
Author Response
- The authors mention the growing trend of incorporating plant-based functional ingredients in meat products. Could they provide more context on the specific drivers behind this trend (e.g., consumer demand, regulatory changes, sustainability concerns) and how their research aligns with these drivers?
Response: Thank you for your feedback. We have added more context on the key factors driving the growing trend of incorporating plant-based functional ingredients into meat products, including consumer demand. We appreciate your valuable feedback. Marked in blue, Lines 39-47.
- The authors could provide a more comprehensive literature review on the specific use of eggplant flour in sausages and related meat products.
Response: Thank you for your comment. The information in the introduction, Lines 65-85, provides a literature review on the specific use of eggplant flour in sausages and related meat products. We appreciate your suggestion.
- While the authors highlight eggplant's antioxidant properties and fiber content, they could elaborate on the potential mechanisms by which these properties could benefit meat products, particularly in terms of oxidative stability, shelf life, and texture.
Response: Thank you for your comment. We have strengthened the discussion section to provide a more in-depth analysis. We appreciate your suggestion. Marked in blue, Lines 264 -266; 280-283; 294-295; 301- 302; 309–311; 316 – 319; 357 – 361; 369 – 373; 384 – 385; 389; 398 – 406; 413 – 423; 428 – 432; 436 – 440; 451 – 452; 454- 458; 466 – 472; 496- 497.
- In the flour preparation section, the authors mention drying the eggplant at 40-45°C to preserve phenolic compounds. Could they provide more details on the rationale behind this temperature range and its efficacy in retaining the desired compounds?
Response: Thank you for your comment. We have added a citation to the text based on results obtained in previous published studies. To provide a broader context, we have expanded the discussion to include the bioactive properties of eggplant rather than focusing solely on phenolic compounds. We appreciate your valuable feedback. Marked in blue, Line 99.
- The authors should clarify the rationale behind the selection of specific eggplant flour concentrations (2%, 3%, 5%, and 9%) in the sausage formulations.
Response: Thank you for your comment. We have added an explanation regarding the selection of the eggplant meal percentage (2% to 9%) in the text. We appreciate your feedback. Marked in blue, Lines 152 – 155.
- Additional information on the sensory evaluation protocol (e.g., sample preparation, serving order, panelist training) would be beneficial.
- The authors mention using a "semi-trained" sensory panel. Could they provide more information on the training protocol, including the number of training sessions, the attributes evaluated, and the reference samples used for calibration?
Response 6 & 7: Thank you for your comment. We have expanded the methodology section to include more details about the sensory evaluation training protocol and the reference samples used. We appreciate your feedback. Marked in blue, Lines 208 – 215.
- The authors should provide a more in-depth discussion of the potential mechanisms behind the observed effects of eggplant flour on the physicochemical, antioxidant, and sensory properties of the sausages.
Response: Thank you for your comment. We have strengthened the discussion section to provide a more in-depth analysis and better support the results. We appreciate your suggestion. Marked in blue, Lines 264 -266; 280-283; 294-295; 301- 302; 309–311; 316 – 319; 357 – 361; 369 – 373; 384 – 385; 389; 398 – 406; 413 – 423; 428 – 432; 436 – 440; 451 – 452; 454- 458; 466 – 472; 496- 497.
- The authors observed a significant enhancement in antioxidant capacity and water retention in sausages with eggplant flour, particularly at 2-3% concentrations. Could they speculate on the potential mechanisms behind these effects and how they might contribute to improved shelf life and textural properties?
Response: Thank you for your comment. We have strengthened the discussion section to provide a more in-depth analysis and better support the results. We appreciate your suggestion. Marked in blue, Lines 357 – 361; 369 – 373; 384 – 385; 389; 398 – 406; 413 – 423; 428 – 432; 436 – 440; 451 – 452; 454- 458; 466 – 472; 496- 497.
- While higher concentrations of eggplant flour (5% and 9%) affected flavor and texture, could the authors provide more specific details on the nature of these sensory changes (e.g., off-flavors, toughness, dryness) and potential strategies to mitigate them?
Response: Thank you for your comment. We have expanded the discussion to include a more detailed argumentation. We appreciate your valuable feedback. Marked in blue, Lines 413 -423.
- The authors mention that eggplant flour proved to be an effective and sustainable option for improving meat product quality. Could they expand on the sustainability aspects of using eggplant flour, such as its environmental impact, resource efficiency, and potential for reducing food waste?
Response: Thank you for your comment. Unfortunately, we have not added further information on the requested topics; however, we have addressed some sustainability aspects in the Introduction and Discussion of Results sections. We appreciate your feedback.
- The discussion could be strengthened by comparing the findings with other relevant studies from the literature.
- In the discussion section, the authors could compare their findings with other studies that have explored similar plant-based functional ingredients in meat products, highlighting the unique contributions of their research.
- The authors could discuss the potential implications of their findings for other meat product categories beyond sausages, such as patties, meatballs, or restructured meat products.
Response 12, 13 & 14: Thank you for your comment. We have strengthened the discussion section to provide a more in-depth analysis and better support the results. We appreciate your suggestion. Marked in blue, Lines 264 -266; 280-283; 294-295; 301- 302; 309–311; 316 – 319; 357 – 361; 369 – 373; 384 – 385; 389; 398 – 406; 413 – 423; 428 – 432; 436 – 440; 451 – 452; 454- 458; 466 – 472; 496- 497.
- It would be beneficial to provide recommendations for future research, such as exploring different processing methods, eggplant varieties, or eggplant flour combinations with other plant-based ingredients.
Response: Thank you for your comment. In the conclusion, we have included recommendations for future research. We appreciate your valuable suggestion. Marked in blue, Lines 505 – 517.

Round 2
Reviewer 1 Report
Comments and Suggestions for Authors
I think author has answered the reviewer’s questions and the manuscript has been improved greatly.
Reviewer 3 Report
Comments and Suggestions for Authors
no more comments